# Advances in Visual Simultaneous Localisation and Mapping Techniques for Autonomous Vehicles: A Review

**DOI:** 10.3390/s22228943

**Published:** 2022-11-18

**Authors:** Jibril Abdullahi Bala, Steve Adetunji Adeshina, Abiodun Musa Aibinu

**Affiliations:** 1Department of Mechatronics Engineering, Federal University of Technology, Minna 920211, Nigeria; 2Department of Computer Engineering, Nile University of Nigeria, Abuja 900001, Nigeria

**Keywords:** autonomous vehicles, computer vision, localisation, mapping, visual SLAM

## Abstract

The recent advancements in Information and Communication Technology (ICT) as well as increasing demand for vehicular safety has led to significant progressions in Autonomous Vehicle (AV) technology. Perception and Localisation are major operations that determine the success of AV development and usage. Therefore, significant research has been carried out to provide AVs with the capabilities to not only sense and understand their surroundings efficiently, but also provide detailed information of the environment in the form of 3D maps. Visual Simultaneous Localisation and Mapping (V-SLAM) has been utilised to enable a vehicle understand its surroundings, map the environment, and identify its position within the area. This paper presents a detailed review of V-SLAM techniques implemented for AV perception and localisation. An overview of SLAM techniques is presented. In addition, an in-depth review is conducted to highlight various V-SLAM schemes, their strengths, and limitations. Challenges associated with V-SLAM deployment and future research directions are also provided in this paper.

## 1. Introduction

Autonomous Vehicles (AVs) have received widespread attention globally due to their low deployment costs, low on-board sensor sizes, and improved technology [1,2,3,4]. AVs have the ability to carry out various driving tasks with minimal or no human supervision. This unique ability of AVs significantly reduces driver workload while increasing driver comfort and efficiency. Additionally, these systems improve vehicle safety by reducing human errors, such as drowsiness, intoxication, fatigue, and loss of concentration, which in turn lead to vehicle accidents [2,5,6,7]. The end-goal of AV development is to create a driverless system that can perform diverse driving tasks [8]. The successful operation of AVs depends on five operational modules which are perception, localisation, decision making, planning, and control [8,9]. While decision making, planning, and control majorly involve problem solving, the vehicle can only perform these tasks based on the input from the perception and localisation modules. Perception, planning, and control are significant challenges in AV security and safety [10]. Understanding the surroundings is critical in a vehicle’s operation [11], and sensing the environment is achieved using a variety of perception technologies such as cameras, lidar, radar, Global Position System (GPS) devices, and ultrasonic sensors [12]. Localisation involves a vehicle estimating its position and orientation within a perceived environment. Robust localisation is another key challenge in AV deployment [13]. Several studies have implemented GPS and Global Navigation Satellite System (GNSS)-based localisation schemes. However, these techniques have proven insufficient in terms of precision, and as such, recent research efforts have focused on the use of lidar and cameras for localisation [13]. A popular technique which utilises cameras and lidar sensors for perception and localisation is Simultaneous Localisation and Mapping (SLAM).

SLAM is method in which an autonomous navigation system obtains 2D or 3D geometric information about its surroundings, which is usually unknown, estimates its pose within that environment, and generates a map of the area [14,15]. SLAM-based systems have been used in a wide range of applications such drones, mobile robots, virtual reality, and augmented reality [14,16]. A variety of sensors have been used in implementing SLAM such as lidar, GPS, and Inertial Measurement Unit (IMU) sensors [14,17]. However, SLAM based on cameras has been widely studied due to the low cost, simpler configuration, low energy consumption, and versatility of cameras [14,16]. This technique, known as Visual SLAM (V-SLAM), uses cameras to estimate an autonomous agent’s position and orientation within an environment, as well as map that environment.

V-SLAM is technique under Visual Odometry (VO) which uses cameras to calculate a body’s change in position over time [18]. V-SLAM has been implemented with different camera types and configurations. Monocular, stereo, and RGB-D cameras have also been successfully used in developing SLAM-based systems [19]. Additionally, these techniques utilises a variety of feature extraction techniques [20] and some researchers have combined traditional V-SLAM schemes with machine and deep learning [21]. Furthermore, the maps generated by SLAM algorithms provide a visual representation of the environment in 2D forms, as in the case of binary occupancy grid maps [22] or 3D forms such as point cloud maps [23].

In this paper, a detailed review of V-SLAM techniques implemented in AVs and mobile robots is presented. The study takes an in-depth look at the techniques, strengths, and limitations of the methods implemented in these systems. The effectiveness and accuracy of current V-SLAM techniques are also highlighted in this study. The main objectives of this paper are as follows:Develop a hierarchy of existing V-SLAM methods with a focus on their respective implementation techniques and perceived advantages over their counterparts;Discuss key characteristics of V-SLAM techniques in literature and highlight their advantages and limitations;Perform a comparative analysis of recent V-SLAM technologies and identify their strengths and shortcomings;Identify open issues and propose future research directions in V-SLAM schemes for AVs.

The rest of this paper is organised as follows. Section 2 provides a background on SLAM with focus on the different types of SLAM algorithms and their implementations. An in-depth look at V-SLAM schemes, their types, architectures, and applications are presented in Section 3. Proposed future directions in the area of V-SLAM research are presented in Section 4 while the conclusion is provided in Section 5.

## 2. Simultaneous Localisation and Mapping (SLAM)

Simultaneous Localisation and Mapping (SLAM) is a major problem in the field of robotics and autonomous navigation systems. The solution to the SLAM problem enables a mobile robot or AV to create a map of its environment while simultaneously keeping track of its pose within the same environment [24,25]. In the last three decades, significant achievements have been recorded in the field of SLAM [16]. SLAM has been implemented in ground [26], aerial [27], and underwater [28] systems.

The SLAM problem has been solved using a variety of approaches. Filter-based algorithms such as Extended Kalman Filter (EKF) and Particle Filter (PF) have been used in SLAM systems. Estimation theoretic approaches such as Bayesian estimation have also been implemented to solve the SLAM problem [24]. For instance, in [29], an optimized self-localization technique for SLAM was presented. This method was developed for dynamic scenes using hypothesis density filters. The novel approach, called GEM-SLAM, achieved major improvements over three benchmark algorithms namely, the Rao Blackwellized Probability Hypothesis Density (RB-PHD) filter, the Single Cluster Probability Hypothesis Density (SC-PHD) filter, and the Factored Solution to SLAM (FastSLAM) algorithm.

### 2.1. SLAM Techniques

SLAM techniques can be categorised into two groups: the Filter-based SLAM and the Graph-based SLAM [29]. The graph-based techniques depend on non-linear optimization techniques and thus, the method used must ensure convergence to global minima. Additionally, pose graph optimisation is susceptible to accumulated errors. Loop closure is used to detect a previously visited location and recalibrates the system’s trajectory [29]. Other graph-based techniques include Occupancy Grid-based SLAM [30], Graph SLAM [30], and Incremental Smoothing and Mapping (iSAM) [31,32].

Because of the limitations of graph-based approaches, such as convergence to local minima and cumulative errors, filter-based techniques have been investigated as possible alternatives to graph-based techniques. The EKF is one of the earliest methods for solving SLAM and was proposed by [33]. A clutter resistant SLAM algorithm for autonomous guided vehicles in dynamic industrial environments was presented in [24]. This clutter-resistant technique uses point features generated from reflectors and line features to improve SLAM robustness. The technique proved to be accurate and efficient by keeping the localisation error within 19 mm and 31 mm for the X-axis and Y-axis, respectively.

The Unscented Kalman Filter (UKF) is related to the class of Linear Regression Kalman Filters. This technique is widely used in SLAM systems to linearize random variable non-linear functions using linear regression [34]. A localisation technique for Wireless Sensor Networks (WSNs) based on UKF and PF algorithms was presented in [34]. The authors modified a Kalman Filter based on UKF and PF localisation. This proposed technique showed a variety of applications such as target tracking and robot localisation based on the simulation results.

The PF is also another filter-based method used in SLAM. The PF samples multiple possibilities (particles) of the observer and filters them based on the likelihood of them having the correct pose [35,36]. A fast algorithm of SLAM for mobile robots based on ball PF was presented in [26]. This technique was a modification of the box PF and the firefly algorithm is used to maintain the diversity of ball particles. This process increased the consistency of the pose estimation and the results demonstrated the superiority in performance of the technique. In addition, ref. [37] presented an embedded PF SLAM implementation using a cost effective platform. In this study, a Genetic Algorithm (GA)-based approach is implemented for calibration and to prevent overfitting. This process provided more generalizable results, robustness, and improved performance. Furthermore, a PF SLAM using differential drive mobile robot was presented in [38]. The study implemented a PF-based SLAM technique on a mobile robot called e-puck. This study observed that the number of particles used had an effect of the performance of the algorithm. Also, ref. [39] developed a stratified PF SLAM based on monocular cameras. This study used a sample weighting algorithm to stratify the particles and was implemented for an Unmanned Aerial Vehicle (UAV). The results were compared to different sample weighting approaches and exhibited improved robustness and accuracy.

In addition, the Rao-Blackwellized Particle Filter (RBPF) is used for robust estimation in SLAM [29]. For instance, ref. [40] presented a study which improved grid-based SLAM with RBPF by adaptive proposals and selective resampling. This technique computed an accurate proposal distribution which considers both the movement of the robot and its most recent observation. The results exhibited the advantage of this technique over other previous approaches. Another popular filter-based technique used in SLAM is the Factored Solution to Simultaneous Localization and Mapping (FastSLAM) which is based on the RBPF [41,42]. The FastSLAM algorithm was later improved by Montemerlo [43,44]. A hierarchy of the types of SLAM techniques is presented in Figure 1.

### 2.2. Sensors Used in SLAM

Autonomous navigation systems use a variety of sensors to perceive the environment and localise themselves. These sensors include lidar, sonar, a Radio Frequency Identification (RFID), Global Positioning System (GPS), Inertial Measurement Units (IMU), and vision sensors [17]. For instance, ref. [45] evaluated the exact flow of particles used for state estimations in unmanned aerial system navigation. The technique utilised sensor information obtained from IMU and GPS sensors and implemented a novel Bayesian filtering SLAM technique for aerial environments. The results proved the effectiveness of this technique over existing methods in terms of speed of convergence and accuracy. Furthermore, ref. [46] presented a PF-based landmark mapping for SLAM of mobile robots. This technique was based on a Radio Frequency Identification (RFID) system and used two separate filters to estimate position and orientation of the robot and integrated circuit tags. The experimental results validated the technique and showed its computational efficiency.

SLAM based on lidar sensors is a popular research area since the sensor data is not affected by illumination changes unlike cameras [47]. This technique is considered to be an accurate and effective method for autonomous agents to create a map and localise themselves [25]. A conic feature-based SLAM technique was developed for open environments in [25]. The technique utilised 2D lidar and a defined conic feature-based parametric approach. The experimental results proved the accuracy of the algorithm in open environments. In addition, ref. [48] benchmarked PF algorithms for efficient Velodyne-based vehicle localisation. Due to the large amount of information obtained by the 3D Velodyne lidar sensor, an analysis was performed to ascertain the how many points were required to achieve optimal efficiency and positioning accuracy. The study conclude that an initial density of two particles per square metre were required to achieve total convergence.

Additionally, Global Navigation Satellite System (GNSS) has been successfully used to implement localisation techniques in SLAM. GNSS provides multiple positioning solutions such as precise point positioning, single point position, and Real-time Kinematic (RTK). GNSS is low cost and works well in an open sky environment [47]. A continuous positioning algorithm based on RTK and Visual Inertial SLAM (VI-SLAM) was presented in [49]. In this study, VI-SLAM was used to complement the limitations of RTK such as blockage of satellite signals by trees and buildings. The results showed the effectiveness of the technique to provide continuous positioning solutions. Furthermore, sonar sensors have been successfully implemented in SLAM systems. The sensors’ ability to perform effectively in underwater water scenarios has made it a popular choice. A PF SLAM with 3D evidence grids in underwater environments was presented in [50]. This study utilised the RBPF in a flooded subterranean environment using sonar data, and provided successful results.

Laser scanners have also been implemented to solve SLAM problems due to their ability to provide bearing and range information [51]. A PF-based outdoor robot localization technique using natural features extracted from a laser scanner was presented in [51]. This method applies batch processing which extracts features from full laser scans. The PF is applied to reduce the estimation error associated with EKFs and the technique was verified in real world experiments. Additionally, ref. [52] presented a distributed SLAM technique using improved PF for mobile robot localisation. This method used data obtained from laser sensors and dead reckoning. The system exhibited better tolerance and robustness over Distributed PF SLAM. Table 1 shows the various sensors used in SLAM and highlights their merits and demerits. The sensors outlined have been implemented in various SLAM techniques with varying configurations. Thus, they can be used for both feature-based and graph-based SLAM implementations.

## 3. Visual Simultaneous Localisation and Mapping (V-SLAM)

### 3.1. Background of V-SLAM

V-SLAM is a technique in which an autonomous navigation system uses a vision sensor to construct and update a map of an unknown environment while at the same time keeping track of its position and orientation within that environment [58,59]. In comparison to other sensor data such as lidar, camera data can provide rich and abundant information, which enhances high level operations [60]. The camera route is represented as a set of relative poses in a world reference frame. The environment is represented by landmarks which are objects or keypoint features in each frame. The landmarks remain still in static environments, while the landmarks change their position in the case of dynamic environments [60].

The V-SLAM process can be broken down into the following steps [58]:Acquire, read, and pre-process the data from the camera and other devices;Estimate motion and local map of the scene from adjacent camera frames;Optimise and adjust camera poses;Detect loops to eliminate errors and complete the map.

V-SLAM primarily uses cameras to solve the SLAM problem due to their low cost, versatility, efficiency, and simple configuration [14]. The cameras can be classified into three major types [49], namely: monocular, stereo, and RGB-D cameras. In the monocular camera setup, only one camera is used for visual perception. Significant research has been conducted in monocular V-SLAM with excellent results. MonoSLAM is considered to be the first pure visual monocular SLAM technique. It was presented in 2003 and was capable of pose estimation and feature measurement on a desktop PC [61]. In 2009, PTAM was presented, which comprised of two threads; one for feature point tracking and the other for mapping [62].

ORB-SLAM [63] is a popular and open source monocular SLAM technique which breaks the SLAM process into three threads: tracking, mapping, and loop detection. Numerous monocular SLAM techniques have been built on the ORB-SLAM framework with impressive results. A SLAM map restoration algorithm based on submaps and an undirected connected graph was presented in [14]. This technique is a monocular SLAM method and uses submap connections to reinitialise and rebuild parts of a map in the event of a system trace failure. The method was simulated on a UAV dataset and the results showed that the integrity of the map was preserved in the case of tracking failures. Additionally, semi-direct monocular SLAM with three levels of parallel optimizations was developed in [64]. In this study, a novel SLAM framework is developed. The first half of the technique utilises direct method for camera pose tracking while the second half of the technique uses a feature-based method for refinement of key frames, loop closures, and mapping. This method showed an increase in accuracy and robustness in motion estimation.

In the monocular camera setup, the scale of the system and environment is ambiguous, and the field of view is limited. It is difficult to determine the depth of the scene or estimate distance from the monocular camera setup [49]. Due to these limitations, studies have been conducted to minimise the effect of inability of monocular cameras to estimate scale. For instance, a simple but effective scale estimation technique for monocular visual odometry in road driving scenarios was presented in [65]. The study utilised 3D ground points to estimate the scale of the camera in monocular V-SLAM. The results showed that the method was able to achieve an average translation error of 1.19% on the KITTI dataset, thus, outperforming conventional monocular visual SLAM techniques.

Stereo camera setup utilises two or more cameras to perceive the environment. In this configuration, the cameras can be placed on both the left and right hand sides of the field of vision. With this setup, it is easier to estimate the scale of the environment. Additionally, this configuration allows the use of smartphones in V-SLAM processes since a significant number of these devices come with multiple cameras [49]. ORB-SLAM2 [19] which is an extension of ORB-SLAM suitable for stereo and RGB-D cameras caters for the limitations associated with monocular cameras. In [66], a persistent map saving technique was developed for visual localisation for AVs. This method was an extension of ORB-SLAM2 and utilised stereo vision to develop a map saving feature for the technique. Experimental results show that the method was capable of keeping the relative translation error of the localisation under 1%. Similarly, a robust stereo visual SLAM technique for dynamic environments with moving object was developed in [67]. This study is based on the ORB-SLAM2 technique as well, and uses a fundamental matrix to identify dynamic feature points and then uses on the static points to estimate the pose. This in turn improved the localisation accuracy and robustness of the system. The authors in [68] developed an indirect visual simultaneous localization and mapping scheme based on linear models. The model was based on the stereo camera configuration and utilised key-frame insertion and map management to minimise computational redundancy and landmark unreliability. The technique showed improved performance when compared to other techniques such as ORB-SLAM2 and COP-SLAM.

Despite the usefulness of stereo vision in depth estimation, in some devices such as smartphones, the cameras are closely placed to each other making accurate depth estimation beyond 2 metres challenging. RGB-D cameras solve this challenge by using Time of Flight technology to estimate the depth of the environment [49]. KinectFusion proposed the use of Kinect cameras for 3D reconstruction, laying the groundwork for research in RGB-D-based SLAM systems although this method was computationally intensive and slow [58]. ORB-SLAM2 [19] has proven to be an effective an popular choice for RGB-D-based V-SLAM techniques. A pixel-wise motion segmentation for slam in dynamic environments was developed by [69]. The study developed a novel pixel-wise segmentation technique for dynamic elements in a quest to improve accuracy and robustness of RGB-D SLAM. The experimental results showed the novel technique outperformed other state of the art methods for dynamic object removal in SLAM schemes. Additionally, ref. [17] developed a real-time cloud visual simultaneous localization and mapping technique for indoor service robots. The study presents the development of a real-time network-based SLAM technique which aims at minimising computing costs. The results showed the efficiency of the method and its ability to bear the network delay in a Local Area Network (LAN).

### 3.2. V-SLAM Categories

V-SLAM can be categorised into three major classifications, namely: Feature-based methods (Indirect method), Direct methods, and Multi-Sensor methods [14]. These methods are presented in Figure 2.

In the feature based or indirect methods, unique features of the camera frame are extracted and matched with the features of the next frame for tracking. ORB (Oriented FAST and rotated BRIEF) feature extraction [19,63] is a popular choice in V-SLAM. The choice of this technique is based on the fact that ORB features are fast to compute and are invariant to viewpoint. Several V-SLAM studies utilise ORB feature extraction as their method of choice. In [60], a robust visual localization technique in dynamic environments based on sparse motion removal was developed. The authors developed a V-SLAM technique for dynamic environments using Sparse Motion Removal. This method uses ORB feature extraction and evaluates the similarities between two consecutive frames and the difference between the current frame and the reference frame. This process identifies dynamic regions in the SLAM process and the results showed an improvement in the localisation accuracy and robustness in dynamic environments. Similarly, ref. [70] developed a novel integrated framework for V-SLAM in an attempt to bridge the gap between visual servoing and V-SLAM. This study, in a quest to enhance robustness and efficiency, developed a novel integrated framework which uses V-SLAM to stabilise servo tasks. The technique also removes feature points of moving objects to enhance operability in dynamic environments. Furthermore, ref. [71] developed MGC-VSLAM which is a meshing-based and geometric constraint V-SLAM technique for dynamic indoor environments. The study improves the accuracy and robustness of the ORB-SLAM2 algorithm by utilising a novel meshing-based uniform distribution approach. Additionally, a modified geometric constraint method is used to filter out the dynamic features. The results showed the technique improved the positioning accuracy of ORB-SLAM2 in highly dynamic surroundings.

ORB feature matching can be adversely affected by illumination changes and weak texture, leading some researchers to explore other feature extraction techniques such as lines and planes [14,72]. For instance, a SLAM system based on RGBD image and point-line feature was developed in [72]. This technique uses RGB-D images and point-line features to improve the accuracy and reliability of SLAM estimations. The results showed an improvement in accuracy of pose estimation and map reconstruction. Additionally, ref. [73] developed DT-SLAM which is a dynamic thresholding-based corner point extraction technique in SLAM systems. The study presents a dynamic self-adaptive threshold technique for corner point detection in a quest to improve SLAM localisation performance. This technique improved upon the ORB corner point extraction in FAST which in turn improved the localisation performance. Furthermore, ref. [74] developed PL-GM which is an RGB-D SLAM with a novel 2D and 3D geometric constraint model of point and line features. The author presented a novel geometric constraint model to utilise both 2D and 3D information within points and lines. The results showed a comparable performance to contemporary SLAM techniques based on point and line features. A line flow-based SLAM technique was presented in [75]. This study uses a line flow to encode line segment observations along the temporal dimension. This technique showed higher efficiency and localisation accuracy. The experimental results showed good localisation and mapping abilities in challenging scenarios.

The direct methods cater for large time consumption associated with feature extraction and matching. Additionally, since the extracted features only represent a small part of the overall image, the direct methods use global pixel information. These techniques work on the assumption that the image intensity should be consistent in corresponding spatial points of neighbouring frames [14]. For instance, ref. [76] presented a semi-direct visual odometry which involves extracting the FAST feature points in the camera frame and subsequently evaluating the pose of the camera transformation according to the information around the feature points. Similarly, ref. [77] presented Large-Scale Direct (LSD) SLAM method. This technique computes the depth of semi-dense points with sudden gradient changes such as corners and edges in place of feature extraction. The technique can be run in real-time on a CPU and also has the ability to deal with weaker textures and larger scales. Other direct V-SLAM methods include Direct Tracking and Mapping (DTAM) and Direct Sparse Odometry [78].

Due to the ambiguity associated with monocular V-SLAM, the technique can be combined with other devices such as GNSS and IMU sensors. This is the case in the multi sensor approach [14,49]. These hybrid methods solve the scale ambiguity problem, albeit with additional configuration, computation, and cost requirements [49]. Visual Inertial SLAM (VI-SLAM) is a technique where the camera is fused with an IMU sensor. This method is useful in real-time applications due to its unique properties such as bias correction, automatic estimator initialisation, online extrinsic calibration, and loop detection [49]. A Visual-Inertial RGB-D SLAM with encoders for a differential wheeled robot deep learning-based v-slam techniques was presented in [79]. The study combined encoders with RGB-D cameras to develop a SLAM technique with improved accuracy and a lower Root Mean Square Error (RMSE). Additionally, a stereo visual inertial mapping algorithm for autonomous mobile robot was presented in [80]. Here, the authors presented a stereo visual inertial mapping system in order to enhance the accuracy and consistency of the generated map. The experimental results proved the robustness and effectiveness of the algorithm. Furthermore, ref. [27] developed VPS-SLAM which is a visual planar semantic SLAM technique for aerial robots. The authors presented robust and lightweight visual semantic SLAM technique for aerial robots. This method combines visual inertial odometry techniques and YOLO v2 to extract semantic information and maps an indoor environment, the effectiveness of which was validated with several experiments. A summary of additional works reviewed under the different types of V-SLAM is presented in Table 2.

### 3.3. Deep Learning Applications in V-SLAM

In recent years, V-SLAM has been combined with deep learning techniques. These deep learning models are utilised in various operations in the SLAM pipeline such as depth prediction [90], object detection [91], and semantic segmentation [92]. Due to the significant improvements in accuracy and performance achieved by deep learning techniques, these methods have become a de facto solution for most computer vision tasks such as object detection and classification [28].

In [16], a real-time visual-inertial localization technique using semantic segmentation towards dynamic environments was presented. The study utilises CNN-based semantic segmentation and multi-view geometric constraints to identify and avoid using dynamic object feature points. The results showed the technique had higher localisation accuracy and robustness when compared to other state of the art SLAM techniques. In addition, ref. [93] presented RDMO-SLAM which is a real-time V-SLAM for dynamic environments using semantic label prediction with optical flow. This technique, which is an extension of [92], is a novel semantic visual SLAM technique based on Mask R-CNN and PWC-Net and was developed to improve the tracking and real time performance of the SLAM algorithm. The results showed an improvement in real-time performance while retaining robust tracking capabilities.

Furthermore, ref. [58] developed a mobile robot visual slam system with enhanced semantics segmentation. The technique was able to improve tracking and speed using camera and encoder data. This was achieved using enhanced semantic segmentation. In [94], the authors present a Visual SLAM method based on semantic segmentation and geometric constraints. This method was suitable for dynamic indoor environments.The test results on the Oxford and TUM datasets demonstrate that the modified approach improves the consistency of feature points extracted by 56.3%. Visual SLAM placement error is decreased by 68.8% on average, and the produced semantic map contains rich semantic information with less redundancy. However, the accuracy and computation time could be improved upon. A summary of additional works reviewed in the area of deep learning-based V-SLAM techniques is presented in Table 3.

## 4. Challenges and Open Issues in V-SLAM

Based on the reviewed works, a number of challenges associated with V-SLAM deployment were identified. These salient limitations can serve as open issues for researchers and provide a focus for future research directions. These challenges are presented as follows:(a)**Reliability in Outdoor Environments**: There is room for improvement in the reliability of V-SLAM implementations, especially in outdoor environments. The ineffectiveness of lidar and radar sensors in extreme weather conditions coupled with their high cost makes them [13,102,103] unsuitable for outdoor conditions. Additionally, despite the high precision and strong anti-interference ability of laser scans, they provide no semantic information about the environment [58]. The use of V-SLAM techniques cater for these limitations, however, these methods are susceptible to unpredictable and uncontrollable environmental conditions such as illumination changes [104].(b)**Operability in Dynamic Scenes**: The traditional SLAM and V-SLAM techniques assume a static environment, which is not always the case. V-SLAM techniques based on static scenes fail when deployed to dynamic environments [57]. The dynamic environment comprises of moving objects which needs to be taken into account in localisation and mapping operations. In the case of ORB-SLAM, for instance, it is not possible to determine if the extracted feature points are from static or dynamic objects [58]. Although significant research has been carried out in the area of object detection [91] and semantic segmentation [55], V-SLAM implementations in highly dynamic sceneries such as road networks and highways have not been exhaustively explored. Autonomous systems still need to fully comprehend dynamic scenarios and cope with dynamic objects [55,99].(c)**Robustness in Challenging Scenes**: V-SLAM techniques need to be robust enough to handle various scenarios. V-SLAM systems have the tendency to fail in situations involving fast motion [57,99]. Additionally, conventional V-SLAM systems rely on stable visual landmarks, which makes implementation difficult [104]. Therefore, achieving robust performance in challenging sceneries is paramount to the success of V-SLAM techniques [105].(d)**Real-time Deployment**: Deployment onto embedded hardware is another open issue for V-SLAM implementations [106]. Existing techniques have high computational requirements and slow real-time performance, thus, resulting in high deployment costs. With a high demand of unmanned systems for deployment in sectors such as Agriculture, Oil and Gas, and Military, the need arises for V-SLAM methods can be deployment onto microcontrollers and microcomputer systems.(e)**Control Scheme for Navigation**: Majority of the reviewed works lack an effective control technique for navigation based on the V-SLAM output. Considering path planning and control are major modules in AV deployment [8,9], there is a need for an effective control mechanism to navigate the AV in relation to the perceived environment. This will significantly contribute to the advancement towards fully autonomous vehicles.

## 5. Conclusions

A detailed review of V-SLAM approaches used for AV perception and localisation was presented in this paper. In addition, an in-depth study of state-of-the-art methodologies in V-SLAM was undertaken, as well as a thorough examination of these methods.

An overview on SLAM methods was provided. The classic SLAM methodology is divided into two types: filter-based and graph-based approaches. Filter-based approaches are preferable for solving the SLAM problem because they are not prone to accumulation errors and do not suffer from convergence to local minima. Furthermore, the various types of sensors utilized in the traditional SLAM technique were addressed, with an emphasis on their benefits and drawbacks. The various camera configurations (monocular, stereo, and RGB-D) and V-SLAM algorithms were also investigated. ORB-SLAM has been recognized as a popular alternative among researchers, particularly as a foundation for developing modified V-SLAM approaches.

Furthermore, a thorough examination was carried out to highlight various V-SLAM systems, their strengths and weaknesses. It was observed that Deep learning models have been used to include tasks such as object identification, semantic segmentation, and depth prediction into SLAM as a result of developments in deep learning. This paper also discussed the challenges of V-SLAM implementation and future research directions. The primary research gaps highlighted were dependability in outdoor environments, operability in dynamic scenarios, robustness in demanding circumstances, real-time deployment, and navigation control methods.

## Figures and Tables

**Figure 1 sensors-22-08943-f001:**
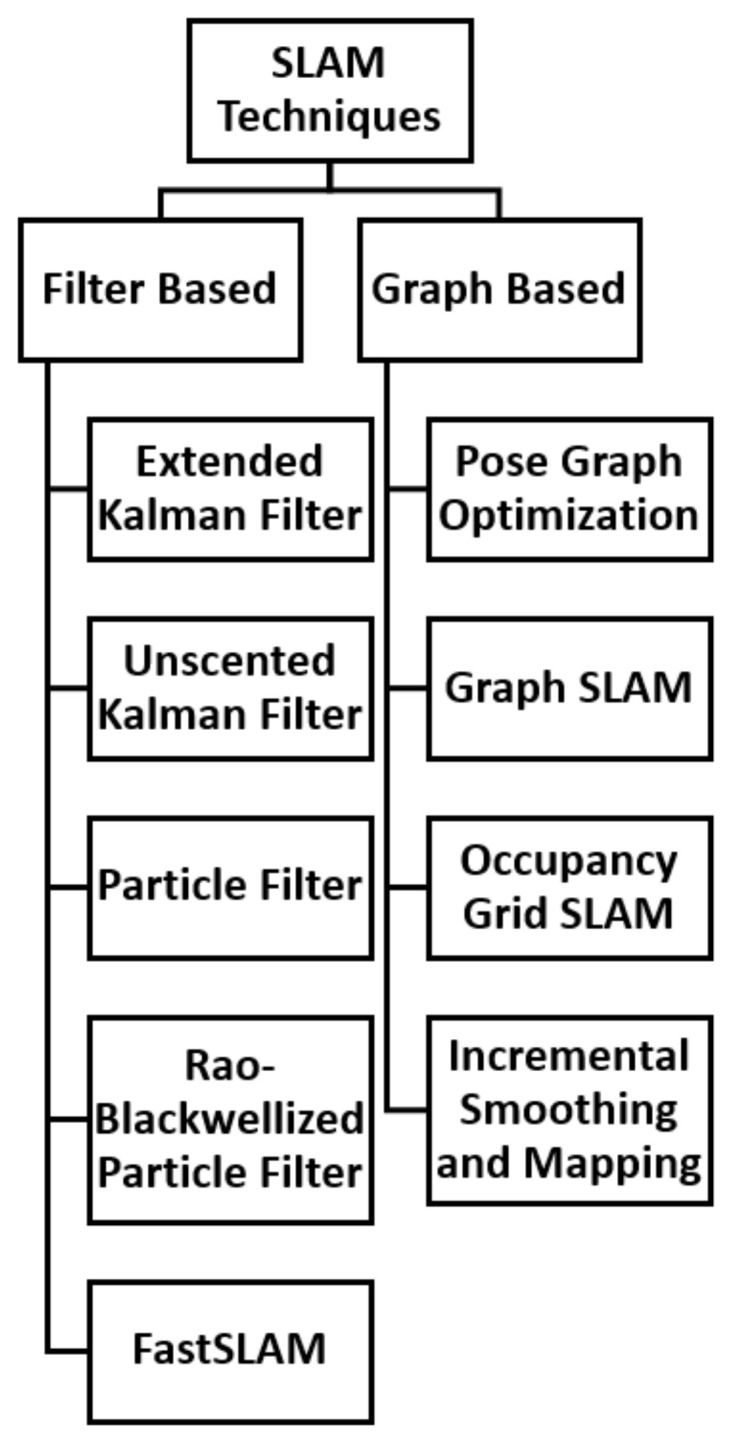
Hierarchy of SLAM Techniques.

**Figure 2 sensors-22-08943-f002:**
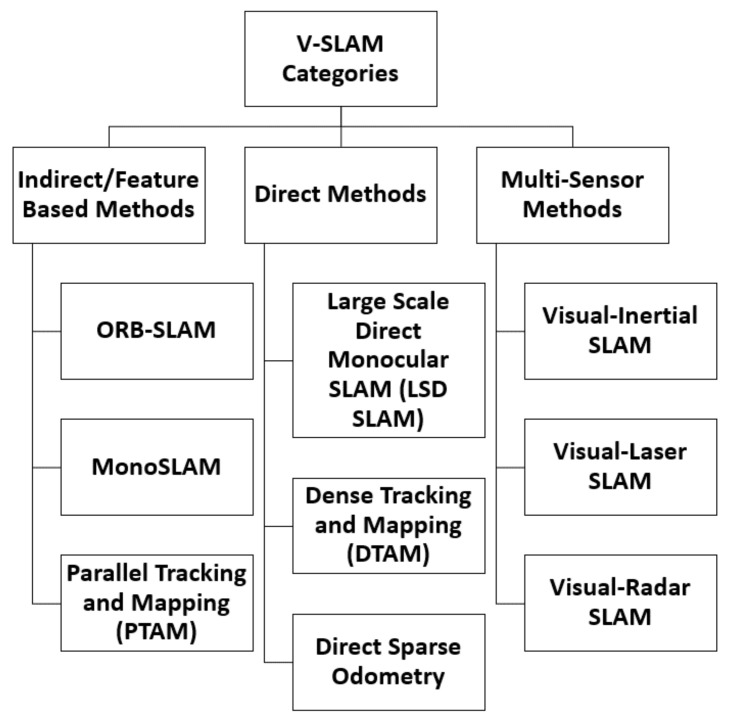
V-SLAM Categories.

**Table 1 sensors-22-08943-t001:** Overview of SLAM Sensors.

Sensor	Advantages	Limitations
Lidar	Can provide 3D and 2D information [25]. Can be used as a standalone SLAM technique [25].	High Cost [49]. Noisy Data [53].
Radar	Provides accurate range information [54].	High Cost [49].
Sonar	Can be used effectively in underwater scenarios [50].	High memory requirements [50]. High Cost [49].
Camera	Low cost, versatile, and simple configuration [17]. Can extract semantic information [55].	Monocular cameras provide no scale information [16]. Performance varies in scenes prone to illumination changes [56].
Inertial Measurement Unit	Can measure angular velocity and acceleration [16]. Serves as complementary sensor to cameras [49]. Good relocalisation capabilities [16].	Difficult to implement as a stand-alone SLAM sensor. Performance unsatisfactory in dynamic environments since it cannot extract semantic information [16].
Wheel Encoders	Not susceptible to drift with temperature [57]. Requires no initialisation [57].	Only suitable for 2D motion [57].
Laser	Can provide accurate range information [24].	Susceptible to errors in reflective environments [24]. High cost [49].
GPS	- Provides accurate location information [49].	Susceptible to failure in environments where coverage is blocked by trees or buildings [49]
GNSS	Provides multiple positioning solutions [49].	Fails in environments where coverage is blocked by trees or buildings [49].

**Table 2 sensors-22-08943-t002:** Related Works in V-SLAM Research.

Ref.	Work	Observations
[57]	Uses measurements of RGB-D camera and encoder to produce robot poses and octo-map, relies on CPU not GPU, works in both static and dynamic indoor environments.	Not designed for outdoor environments, problem of wheel slipping causes inaccuracy, inability to track dynamic objects.
[81]	Technique proved agreement between system pose estimates and ground truth.	Performance on dynamic outdoor environment could not be determined, and depth camera in Kinect blinded by sunlight during daytime.
[49]	Uses Visual-Inertial SLAM to complement limitations of RTK such as blockage of satellite signals due to buildings and trees. This was achieved with a common smartphone instead of extra specialised devices.	Further work required to evaluate the model more accurately, not designed for dynamic outdoor environments. RTK systems are reliant on further infrastructure which comes with additional costs according to [66].
[82]	System uses pre-existing map and compares obtained images to evaluate user’s position within the map.	Use of pre-existing map not suitable for dynamic outdoor environments.
[83]	The study presents the development of a new processing chain based on V-SAM for UAVs.	Data processing performance in real time is low, and the technique focuses on aerial motion and thus, no information was provided for ground movement.
[84]	In this study, the authors developed a semantic depth filter for RGB-D SLAM operations making it more accurate in dynamic environments.	Simulated using TUM dataset and thus performance on dynamic outdoor environment could not be determined.
[85]	The study presents novel panoramic Visual Inertial SLAM which utilises a wheel encoder to achieve improved robustness and localisation accuracy.	Simulated using University of Michigan North Campus Long-Term Vision and LiDAR Dataset (NCLT) dataset and thus performance on dynamic outdoor environment could not be determined.
[86]	The study presents a SLAM algorithm coupled with wheel encoder measures to enhance localisation. A low cost map was generated to enhance speed and memory efficiency.	Despite its ability to handle tracking in dynamic scenarios, the system only considered indoor settings and no information on outdoor performance was provided.
[87]	The study uses wheel odometer measurements and monocular camera to develop a Visual Inertial Odometry model coupled with non-holonomic constraints.	Simulated using KITTI and KAIST Complex Urban datasets and thus, real time performance on dynamic outdoor environment could not be accurately determined.
[88]	This article presents Integrated Visual Odometry with a Stereo Camera (IVO-S), a unique low-cost underwater visual navigation approach. Unlike pure visual odometry, the suggested approach combines data from inertial sensors and a sonar to function in context-sparse situations.	The suggested approach performs effectively in underwater sparse-feature settings with high precision, but existing visual slams or odometries, like as ORB-SLAM2 and OKVIS, do not. However, the technique does not include loop closure detection and map reconstruction operations.
[89]	This research presents a real-time and resilient point-line based monocular visual inertial SLAM (VINS) system for smart city mobility robots heading towards 6G. EDLines with adaptive gamma correction are used to extract a higher proportion of long line features among all extracted line features faster.	The experimental findings reveal that the VINS system outperforms other sophisticated systems in terms of localization accuracy, and robustness in challenging situations. However, the performance in outdoor scenes could not be accurately determined since the model was not deployed in outdoor settings.

**Table 3 sensors-22-08943-t003:** Deep Learning-Based V-SLAM Techniques.

Ref.	Work	Observations
[20]	This study presents a SLAM technique that uses objects and walls as elements of the environment model. The objects are identified using YOLO v3 technique. The system exhibited better performance than the RGB-D SLAM and a comparable performance to ORB SLAM.	The technique was tested in a static indoor environment and thus, the performance on a dynamic outdoor environment could not be determined.
[95]	In this study, a semantic filter-based faster R-CNN is utilised to solve fundamental matrix calculations in ORB SLAM. This method reduced the trajectory error, number of low quality feature correspondences, and position error.	Simulated sing KITTI ad ETH datasets and thus performance on dynamic outdoor environment could not be determined.
[55]	Developed a novel RGB-D SLAM method combined with deep learning in order to decrease impact of moving objects in the estimation of camera pose. This was achieved using semantic segmentation and multi-view geometry.	Real time performance needs to be improved, not designed for outdoor environments.
[96]	Technique combined ORB-SLAM2 and PSPNet-based semantic segmentation to identify and eliminate dynamics points. This reduced the trajectory and pose errors.	Simulated using the TUM RGB-D dataset and thus performance on dynamic outdoor environments could not be determined.
[97]	Here, a Decoder-Encoder Model (DEM) was developed which uses CNNs to improve depth estimation performance. Additionally, a loss function was developed to enhance the training of the DEM.	Simulated using indoor NYU-Depth-v2 and outdoor KITTI datasets and thus performance on dynamic outdoor environment could not be determined.
[98]	In this study, YOLO v3 was used to provide semantic information in order to distinguish edge features and reduce the effect of unstable features. This process improved the positioning accuracy of the system.	Simulated using public TUM RGB-D dataset and thus performance on dynamic outdoor environment could not be determined.
[16]	The study utilises CNN-based semantic segmentation and multi-view geometric constraints to identify and avoid using dynamic object feature points.	Simulated using ADVIO dataset and thus performance on dynamic outdoor environment could not be determined.
[99]	The study presents a dynamic point detection and rejection algorithm centred on neural network-based semantic segmentation. This eliminates dynamic object interference during pose estimation.	The technique was simulated on the EuRoC dataset and collected underground images tunnel. However, the real-time performance could not be evaluated on dynamic ground outdoor environments.
[100]	Presented in this study was a novel Visual Place Recognition technique capable of operating under changing viewpoint and appearance conditions. The system avoids the use of CNN which has high computational requirements.	The system was simulated on various public VPR datasets but focused mainly on static environments.
[101]	A deep learning-based real-time visual SLAM technique is proposed in this work. A parallel semantic thread is created using the lightweight object detection network YOLOv5s to obtain semantic information in the scene more quickly.	The experimental findings suggest that the system improves in terms of accuracy as well as real-time performance. However, for practicality, the map generating process and computation speed need to be improved.

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
