# Peer review of "Advances in Visual Simultaneous Localisation and Mapping Techniques for Autonomous Vehicles: A Review"

_sensors, 2022, doi:10.3390/s22228943_

Round 1

Reviewer 1 Report

The authors have done a good job in presenting a paper in an emerging area. The organization of the paper and flow is perfect. I have only one suggestion. As the paper is a review paper,the citations from the year 2022 is minimum.I suggest to include few relevant papers of 2022 if available in the revised manuscript.

Author Response

Dear Reveiwer, I hope this email meets you well. On behalf of myself and my co-authors, I want to express our appreciation for your review. Your comments have added quality to the work. Please see the attachment for a point to point response to your comments. Thank you.

Reviewer 2 Report

Overall, paper is acceptable to the journal Sensors.

There are comments that I want to point out and typos.

1) Line 62-65 : This is only a list. Hope to add more information to decide what is important in each method.

2) Line 85 : What are those `benchmarks?'

3) Line 94 : Hope to name reason(s) for the description.

4) Figure 1 : Authors may want to improve the quality (resolution).

5) Table 1 : It is required to mention the criteria for the classification.

Finally, it could be much better to improve the conclusion. Currently it looks just a summary of the paper.

Author Response

(The authors gave the same response as above.)
